# Bacterial survival in microscopic surface wetness

**Maor Grinberg[†], Tomer Orevi[†], Shifra Steinberg, Nadav Kashtan***

Department of Plant Pathology and Microbiology, Robert H. Smith Faculty of Agriculture, Food, and Environment, Hebrew University, Rehovot, Israel

**Abstract** Plant leaves constitute a huge microbial habitat of global importance. How microorganisms survive the dry daytime on leaves and avoid desiccation is not well understood. There is evidence that microscopic surface wetness in the form of thin films and micrometer-sized droplets, invisible to the naked eye, persists on leaves during daytime due to deliquescence – the absorption of water until dissolution – of hygroscopic aerosols. Here, we study how such microscopic wetness affects cell survival. We show that, on surfaces drying under moderate humidity, stable microdroplets form around bacterial aggregates due to capillary pinning and deliquescence. Notably, droplet-size increases with aggregate-size, and cell survival is higher the larger the droplet. This phenomenon was observed for 13 bacterial species, two of which – *Pseudomonas fluorescens* and *P. putida* – were studied in depth. Microdroplet formation around aggregates is likely key to bacterial survival in a variety of unsaturated microbial habitats, including leaf surfaces.

DOI: https://doi.org/10.7554/eLife.48508.001

## Introduction

The phyllosphere – the aerial parts of plants – is a vast microbial habitat that is home to diverse microbial communities (*Lindow and Brandl, 2003*; *Lindow and Leveau, 2002*; *Vorholt, 2012*; *Vacher et al., 2016*; *Leveau, 2015*; *Bringel and CouÃce, 2015*). These communities, dominated by bacteria, play a major role in the function and health of their host plant, and take part in global biogeochemical cycles. Hydration conditions on plant leaf surfaces vary considerably over the diurnal cycle, typically with wet nights and dry days (*Beattie, 2011*; *Brewer and Smith, 1997*; *Magarey et al., 2005*; *Klemm et al., 2002*). An open question is how bacteria survive the dry daytime on leaves and avoid desiccation.

While leaf surfaces may appear to be completely dry during the day, there is increasing evidence that they are frequently covered by thin liquid films or micrometer-sized droplets that are invisible to the naked eye (*Burkhardt and Hunsche, 2013*; *Burkhardt and Eiden, 1994*; *Burkhardt et al., 2001*) (*Figure 1A*). This microscopic wetness results, in large part, from the deliquescence of hygroscopic particles that absorb moisture until they dissolve in the absorbed water and form a solution. One ubiquitous source of deliquescent compounds on plant leaf surfaces is aerosols (*Pöschl, 2005*; *Tang and Munkelwitz, 1993*; *Tang, 1979*). Notably, during the day, the relative humidity (RH) in the boundary layer close to the leaf surface is typically higher than that in the surrounding air, due to transpiration through open stomata. Thus, in many cases, the RH is above the deliquescent point, leading to the formation of highly concentrated solutions in the form of thin films (<a few µm) and microscopic droplets (*Burkhardt and Hunsche, 2013*). The phenomenon of deliquescence-associated microscopic surface wetness is under-studied, and little is known about its impact on microbial ecology of the phyllosphere and on its contribution to desiccation avoidance and cell survival during the dry daytime.

*For correspondence:
nadav.kashtan@mail.huji.ac.il

[†]These authors contributed equally to this work

**Competing interests:** The authors declare that no competing interests exist.

**eLife digest** A single plant leaf can be home to about 10 million bacteria and other microbes. These microscopic organisms are part of a larger community of microbes – the microbiome – that plays an important role in the life and health of their plant host. Like all other organisms, bacteria need water to survive, but the surfaces of leaves experience daily changes in moisture, tending to be much wetter at night than during the day.

While the surfaces of leaves often appear dry during the day, previous studies suggest they may actually be covered by thin films or tiny droplets of fluid that are invisible to the naked eye. This microscopic wetness forms because hygroscopic particles such as aerosols, which tend to absorb moisture from the air, are common on the leaf surface. These molecules absorb water until they become dissolved in it, leaving behind a concentrated solution (a process known as deliquescence). However, it is not clear if this microscopic wetness can protect bacteria from drying out.

Here, Grinberg, Orevi et al. investigated how bacteria, including several species that are commonly found on plants, survived episodes of drying on an artificial surface that produces microscopic wetness. The experiments revealed that as the surfaces dried out, stable microscopic droplets formed around the bacterial cells. The droplets that formed around aggregates of bacterial cells were larger than those that formed around solitary cells. Bacteria inside these droplets can survive longer than 24 hours, and survival rates were much higher in larger droplets.

Further experiments found that 11 other species of bacteria could also survive an episode of drying for over 24 hours if microscopic droplets formed around them. Together, these findings suggest that by organizing themselves into aggregates, bacteria can improve their chance of surviving on the surface of leaves and other environments that are frequently exposed to drying.

These results help explain how microbes avoid drying and survive during the daytime on leaf surfaces. Understanding how microscopic leaf wetness protects the plant microbiome is important because it helps explain how it can be disrupted by agricultural practices and human-made aerosols, information that can be used to better protect plants.

Microscopic surface wetness is likely to occur in many other situations including in the soil, on human and animal skin, and in homes and workplaces. These findings may have broad implications on the way we understand bacterial life on these seemingly dry surfaces, potentially leading to future benefits for human health, agriculture, and nature conservation.

DOI: https://doi.org/10.7554/eLife.48508.002

The microscopic hydration conditions around bacterial cells are expected to significantly affect cell survival in the largest terrestrial microbial habitats – soil, root, and leaf surfaces – that experience recurring wet-dry cycles. Only a few studies have attempted to characterize the microscopic hydration conditions surrounding cells on a drying surface under moderate RH and the involvement of deliquescent substrates in this process. Bacterial survival in deliquescent wetness has mainly been studied in extremely dry deserts (*Davila et al., 2008*; *Davila et al., 2013*) and on Mars analog environments (*Nuding et al., 2017*; *Stevens et al., 2019*). Soft liquid-like substances wrapped around cells, whose formation was suggested to be due to deliquescence of solute components, were reported (*Méndez-Vilas et al., 2011*). Yet, the interplay between droplet formation, bacterial surface colonization, and survival, has not been studied systematically.

Bacterial cells on leaf surfaces are observed in solitary and aggregated forms. The majority of cells are typically found within surface-attached aggregates, that is biofilms (*Monier and Lindow, 2004*; *Morris et al., 1997*). This is consistent with the reported increased survival rate in aggregates under dry conditions on leaves, and poor survival of solitary cells (*Monier and Lindow, 2003*; *Rigano et al., 2007*; *Yu et al., 1999*). The conventional explanation for the increased survival in aggregates is the protective role of the extracellular polymeric substances (EPS), a matrix that acts as a hydrogel (*Chang et al., 2007*; *Or et al., 2007*; *Roberson and Firestone, 1992*; *Ophir and Gutnick, 1994*). Here, we ask if aggregation plays additional roles in protection from desiccation. We hypothesize that the resulting microscale hydration conditions around cells on a drying surface depend on cellular organization (i.e. solitary/aggregated cells and aggregate size) and that the microscale hydration conditions (i.e. droplet size) affect cell survival.

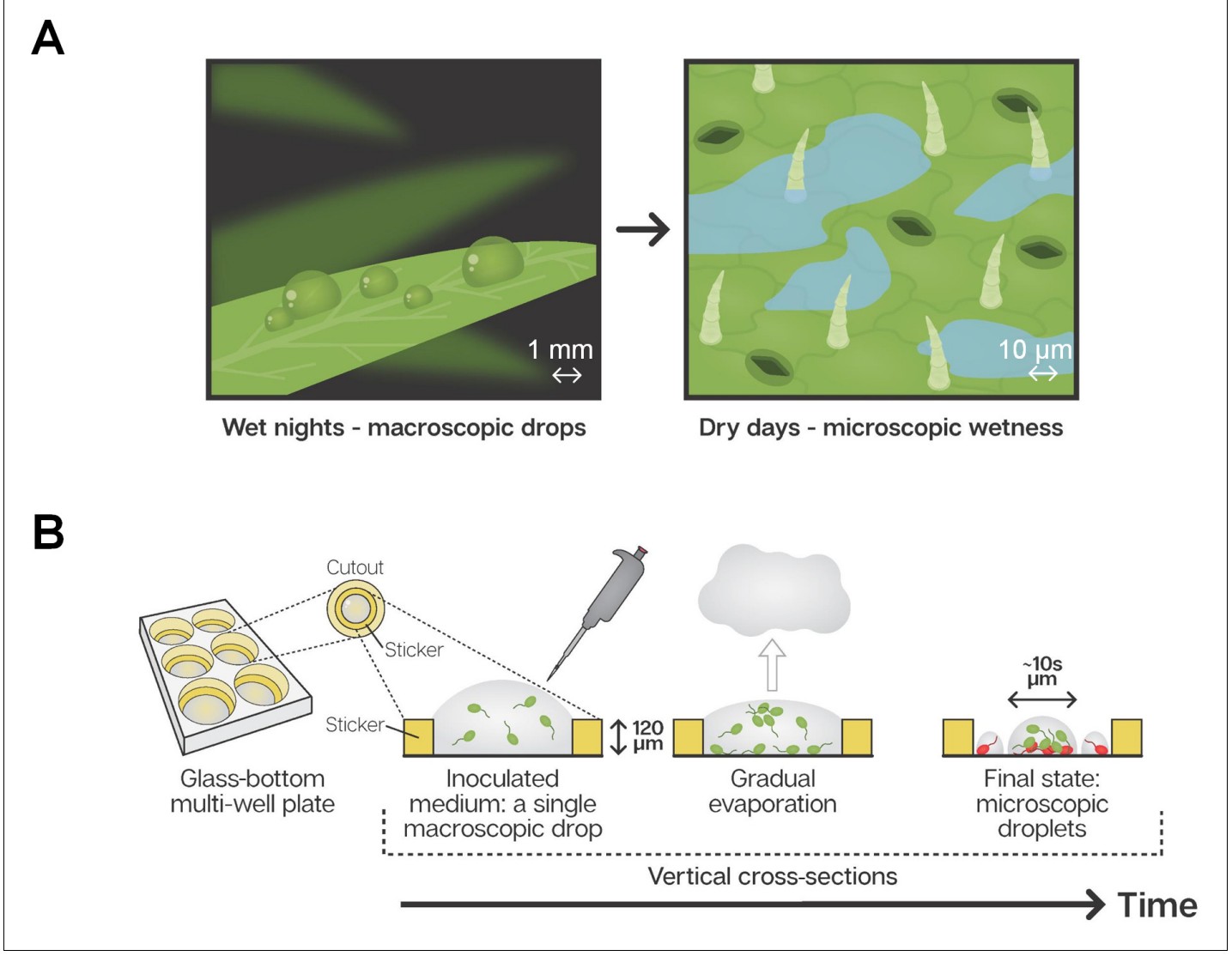

**Figure 1.** Microscopic wetness: Experimental setup. (**A**) Plant leaf surfaces are usually wet at night with visible macroscopic wetness (e.g. dewdrops). During the day, leaf surfaces are typically dry, with microscopic wetness invisible to the naked eye. (**B**) A thin, round sticker is placed in the center of each well in a glass-bottom, multi-well plate. The hollow part of the sticker is loaded with a medium containing suspended bacteria cells. The well-plate is placed under constant temperature, RH, and air circulation. Water gradually evaporates from the medium while bacteria grow, divide, and colonize the surface of the well until the surface becomes macroscopically dry and microscopic surface wetness forms.

DOI: https://doi.org/10.7554/eLife.48508.003

To this end, we designed an experimental system that creates deliquescent microscopic wetness on artificial surfaces. This system conserves some basic important features of natural leaf microscopic wetness while eliminating some of the complexities of studying leaf surfaces directly. The system enabled us to perform a systematic microscopic analysis of the interplay between bacteria's cellular organization on a surface, microscopic wetness, and cell survival on surfaces drying under moderate humidity.

We observed that bacterial cells – aggregates in particular – retained a hydrated micro-environment in the form of stable microscopic droplets (of tens of μm in diameter) while the surface was macroscopically dry. We then quantitatively analyzed the distribution of droplet size, its correlation with aggregate size, and the fraction of live and dead cells in each droplet. The significance of our results is discussed in the context of survival strategies on drying surfaces, microbial ecology of the phyllosphere, and possible relevance to other habitats.

## Results

### Drying experiments on bacteria-colonized surfaces

Studying bacteria in microscopic surface wetness directly on leaves poses a significant technological challenge due to strong auto-fluorescence, surface roughness, and transparency of films and micro-droplets. We therefore constructed a simple experimental system, accessible to microscopy, that enables studying the interplay between bacterial surface colonization, cell survival, and microscopic wetness on artificial surfaces. This system enables capturing microscopic leaf wetness central properties, including contribution of deliquescent substrates, droplet persistence, thickness, and patchiness (*Figure 1B* - see Materials and methods). We studied in depth two model bacterial strains – *Pseudomonas fluorescens* A506 (a leaf surface dweller strain; *Wilson and Lindow, 1993*; *Hagen et al., 2009*) and *P. putida* KT2440 (a soil and root bacterial strain extensively studied under unsaturated hydration conditions; *Nelson et al., 2002*; *Molina, 2000*; *van de Mortel and Halverson, 2004*; *Espinosa-Urgel et al., 2002*). Qualitatively similar results were observed for 16 additional strains (13 bacterial species in total - see Materials and methods). Briefly, bacterial cells were inoculated in diluted M9 minimal media onto hollowed stickers applied to the glass substrate of multi-well plates and placed inside an environmental chamber under constant temperature and RH (28˚C; 70% or 85% RH) (*Figure 1B* - Materials and methods). Results shown here are from 85% RH though 70% RH yielded qualitatively similar results.

### Microscopic droplet formation around bacterial cells and aggregates

At 85% RH, it took about 14 ± 1 hr for the bulk water to evaporate. During this time, for both studied strains, some of the cells attached to the surface and, over time, grew and formed aggregates. Other cells formed cell clusters at the liquid-air interface (pellicles). The rest of the cells remained solitary: either surface-attached, or planktonic. The glass substrate appeared dry to the naked eye after 14 ± 1 hr of incubation. We then examined the surface of the wells under the microscope (see Materials and methods). Remarkably, the surface was covered by stable microscopic droplets, mainly around bacterial aggregates (*Figure 2A–B*). Notably, while solitary cells were surrounded by miniscule droplets (possibly similar to those reported by *Méndez-Vilas et al., 2011*), larger aggregates (of ~100 cells) were surrounded by large droplets measuring tens of μm in diameter. Microscopic wetness was retained around bacterial cells for more than 24 hr, while uncolonized surface areas appeared completely dry.

In order to assess the distribution of droplet size and the correlation between droplet size and aggregate size, we scanned a large area of the surface (~10 mm$^2$) to collect and analyze information on thousands of microdroplets (Materials and methods). We found that droplet size (measured by droplet area) follows a power law distribution with similar exponents for the two studied strains (*Figure 2C*). When droplet size was plotted as a function of area covered by cells within each droplet (as a proxy for cell number - see Materials and methods), a clear positive correlation between cell abundance and droplet size emerged (*Figure 2D*). Experiments using hydrophobic polystyrene substrate rather than glass also yielded qualitatively similar results (*Figure 2—figure supplement 1*).

### The underlying mechanisms of droplet formation

To understand how these microdroplets form, we tested what components of the system were essential to this process. First, we repeated the experiments with fluorescent beads (2 μm in diameter) instead of bacteria. Interestingly, we found that microdroplets formed even around beads (*Figure 2—figure supplement 2*) with a similar droplet-size distribution, as in experiments with bacteria; and a surprisingly similar correlation between the size of the droplet and the number of beads therein (*Figure 2C–D*). In a control experiment without any particulates – bacterial cells or beads – a much smaller number of droplets formed (<1 droplets of >10 μm$^2$ area per mm$^2$, as opposed to >100 droplets of that size in experiments with bacteria). These results indicate that the presence of particles is necessary for droplet formation, whereas biological activity is not. Last, we repeated the beads experiment with pure water instead of M9 medium. This time we did not observe any droplets (*Figure 2—figure supplement 2*), indicating that the solutes control droplet formation and retention through their deliquescent properties.

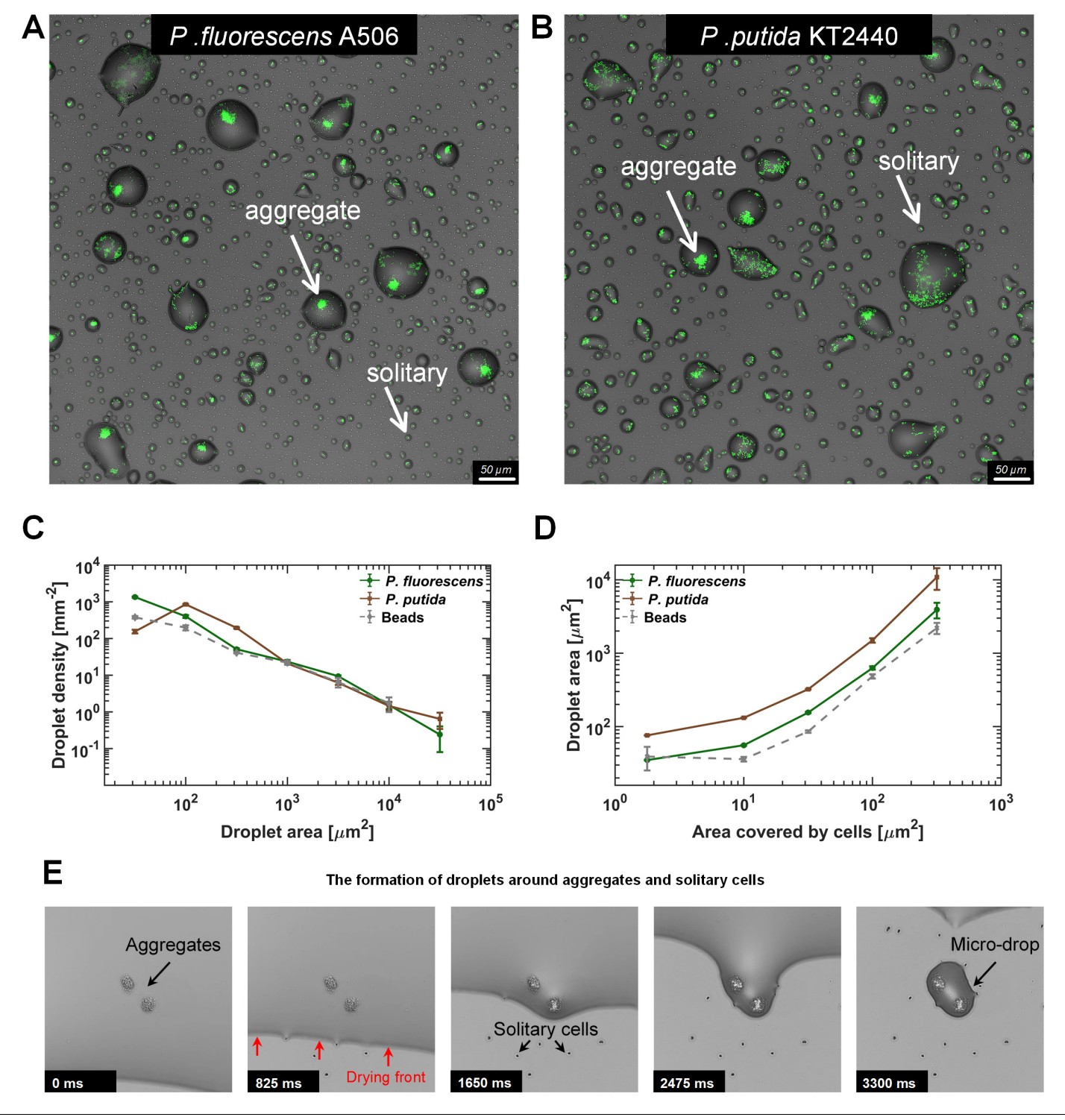

**Figure 2.** Microdroplets form around bacterial cells and aggregates. (A–B) Representative sections of the surface imaged 24 hr after macroscopically dry conditions were established. Bacterial cells (pseudo color in green) that colonized the surface during the wet phase of the experiment are engulfed by microdroplets, while uncolonized portions of the surface appear to be dry. Solitary cells are engulfed by very small microdroplets, while large aggregates are engulfed by larger droplets (white arrows). Images show a 0.66 × 0.66 mm section from an experiment with *P. fluorescens* (A) and *P. putida* (B). (C) Droplet-size distributions at 24 hr: Droplets from both strains show power law distributions with relatively similar exponents (γ = −1.2 ± 0.15 (mean ± SEM) and −1.0 ± 0.45 for *P. fluorescens* and *P. putida*, respectively). (D) Droplet size as a function of cell abundance within the droplet (estimated by area covered by cells): Droplet size increases with cell abundance within the droplet. Error bars in (C) and (D) are standard errors.
*Figure 2 continued on next page*

*Figure 2 continued*

(E) A time-lapse series capturing the formation of microdroplets around bacterial aggregates: The thin (a few µms) liquid receding front clears out from the surface, leaving behind microdroplets whenever it encounters bacterial cells or aggregates (see also *Videos 1–3*).

DOI: https://doi.org/10.7554/eLife.48508.004

The following source data and figure supplements are available for figure 2:

**Source data 1.** Droplet size distributions and their relation to area covered by cells.
DOI: https://doi.org/10.7554/eLife.48508.011
**Figure supplement 1.** The formation of microdroplets on polystyrene substrate.
DOI: https://doi.org/10.7554/eLife.48508.005
**Figure supplement 2.** Drying surface experiment with fluorescent beads (2 µm in diameter).
DOI: https://doi.org/10.7554/eLife.48508.006
**Figure supplement 3.** Estimating the solute concentrations in microdroplets in comparison to standard M9 medium.
DOI: https://doi.org/10.7554/eLife.48508.007
**Figure supplement 3—source data 1.** M9 calibration: relation between concentration factor relative to standard M9 vs intensity.
DOI: https://doi.org/10.7554/eLife.48508.008
**Figure supplement 4.** Estimating NaCl concentrations in microdroplets (medium containing $diH_2O$ + NaCl only).
DOI: https://doi.org/10.7554/eLife.48508.009
**Figure supplement 4—source data 1.** NaCl calibration: relation between concentration [mM] vs intensity.
DOI: https://doi.org/10.7554/eLife.48508.010

To observe the surface's final drying phase, we used time-lapse imaging, enabling us to capture the receding front of the remaining thin liquid layer and the formation of microdroplets. Retention of droplets around aggregates as well as solitary cells, through pinning of the liquid-air interface, is clearly evident (*Figure 2E*, *Videos 1–3*). The cause of this pinning is the strong capillary forces acting on the rough surfaces produced by the presence of particulates (*Bonn et al., 2009*; *Herminghaus et al., 2008*). This phenomenon supports the notion that aggregate sizes (but possibly also other properties) determine droplet size. We note that under our experimental conditions, the droplets were not formed through the wetting 'direction' of a deliquescence process, by which solid salts absorb water until dissolution. Rather, the deliquescent properties of the solutes prevented complete evaporation at RH above the point of deliquescence of the salts mixture. In summary, both particulates and deliquescent solutes are essential for the differential formation

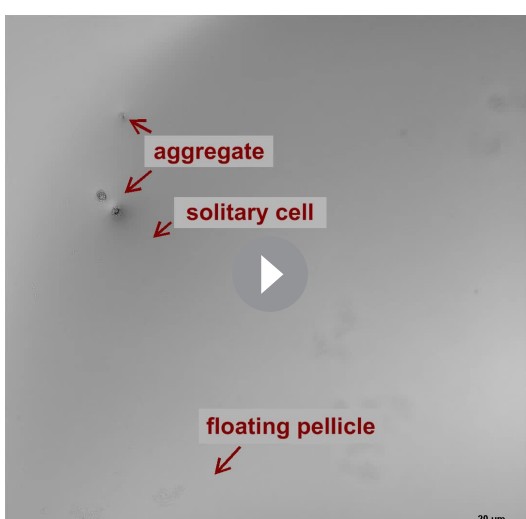

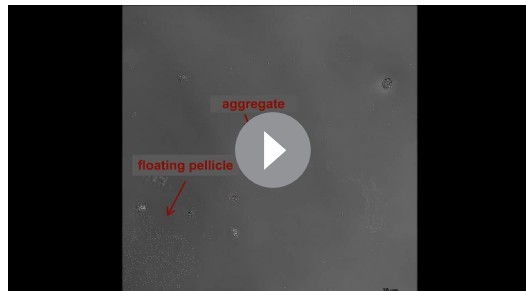

**Video 1.** The formation of microdroplets around bacterial cells. The thin (a few µm thick) liquid's receding front clears out from the surface, leaving behind microdroplets whenever it encounters solitary cells, surface-attached aggregates, or floating pellicles. Videos taken from an experiment with *P. fluorescens* cells. Video was imaged with a 20x objective. The video plays at real-time speed.
DOI: https://doi.org/10.7554/eLife.48508.012

**Video 2.** The formation of microdroplets around bacterial cells. The thin (a few µm thick) liquid's receding front clears out from the surface, leaving behind microdroplets whenever it encounters solitary cells, surface-attached aggregates, or floating pellicles. Videos taken from an experiment with *P. fluorescens* cells. Video was imaged with a 40x objective. The video plays at real-time speed.
DOI: https://doi.org/10.7554/eLife.48508.013

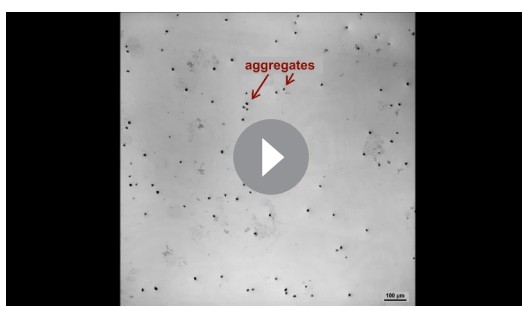

**Video 3.** The formation of microdroplets around bacterial cells. The thin (a few μm thick) liquid's receding front clears out from the surface, leaving behind microdroplets whenever it encounters solitary cells, surface-attached aggregates, or floating pellicles. Videos taken from an experiment with *P. fluorescens* cells. Video was imaged with a 10x objective. The video plays at real-time speed.
DOI: https://doi.org/10.7554/eLife.48508.014

and retention of microscopic wetness around cells and aggregates.

## Microdroplets are highly concentrated solutions

Direct measurements of the solute concentrations within microdroplets constitute a technical challenge. To overcome that challenge, we added a fluorescent dye (Alexa Fluor 647) to the initial M9 medium, as a reporter for the solute concentration induced by evaporation. We compared the fluorescent intensity of dye-labeled microdroplets to a calibration curve built by measuring the intensities of known concentrations of the standard M9 supplemented with Alexa 647 (see Materials and methods, *Figure 2—figure supplement 3*). We found that the microdroplet solution is highly concentrated – as can be expected from deliquescent wetness – and is estimated to be 23.3 ± 3.5 (mean ± SD) more concentrated than a standard M9 (~50 times more concentrated than the diluted 0.5x M9 used in our experiments) (See Materials and methods, *Figure 2—figure supplement 3*). The high estimated mean osmolarity within the droplets (~6.7 Osm/L, see *Supplementary file 1*) likely imposes severe osmotic stress on cells within them. Indeed, growth curves of the two strains (*P. fluorescens* and *P. putida*) in liquid cultures of equivalent concentrated M9 and M9+NaCl media showed delayed or complete growth inhibition (*Figure 3—figure supplement 5*, *Supplementary file 2*). This result accords with the observation that cell divisions within droplets was rarely seen in our experiments (at 85% RH).

### Cell survival rate increases with droplet size

As cells inhabit a heterogeneous landscape of droplets of various sizes, we next asked whether droplet size affects cell survival. We applied a standard bacterial viability assay by adding propidium iodide (PI) to the medium (see Materials and methods). Thus, live cells emit green-yellow fluorescence, while dead cells exhibit red emission (*Figure 3A,B*). The assay's validity was further confirmed by the observation that following further incubation at 95% RH, YFP-expressing cells were dividing (some were even motile), while red cells lacked signs of physiological activity (*Figure 3—figure supplement 1*, *Videos 4* and *5*). Notably, although the overall population distribution along droplet size was strain specific, survival of cells was nearly exclusively restricted to large droplets for both strains (>10³ μm² area; *Figure 3C,D*, Materials and methods). *P. putida* showed higher overall survival than did *P. fluorescens* (16% vs. 7%, 24 hr after drying). We note that the overall survival often varied between experiments, and in some cases *P. fluorescens* had higher survival than did *P. putida*. Importantly, regardless of this stochasticity, common to all experiments was a clear trend for both strains: The fraction of live cells within droplets increases with droplet size (*Figure 3E*). Accordingly, survival probabilities in small droplets (<10² μm² area) were poor (<5%), in contrast to >50% survival of both strains in the largest droplets (>10⁴ μm²).

Next, we sought to understand what the net contribution of droplet size is to cell survival. Analysis of cell survival rates as a function of both aggregate size (which by itself affects survival; *Monier and Lindow, 2003*, cf. *Figure 3—figure supplement 2*) and the size of the droplet they inhabit, shows that for both strains, droplet size strongly affects survival, whereas aggregate size has only a marginal (*P. fluorescens*) or moderate (*P. putida*) effect on survival (*Figure 3F,G*). The relative contribution of each of these two variables was also assessed by a multinomial logistic regression model, giving significantly higher weight to droplet size in comparison to aggregate size, for both strains (*Figure 3—figure supplement 3*).

To further study droplet size effect on survival, we repeated the drying experiment, but inoculated the cells into the drying medium only at a later stage – closer to the macroscopic drying stage

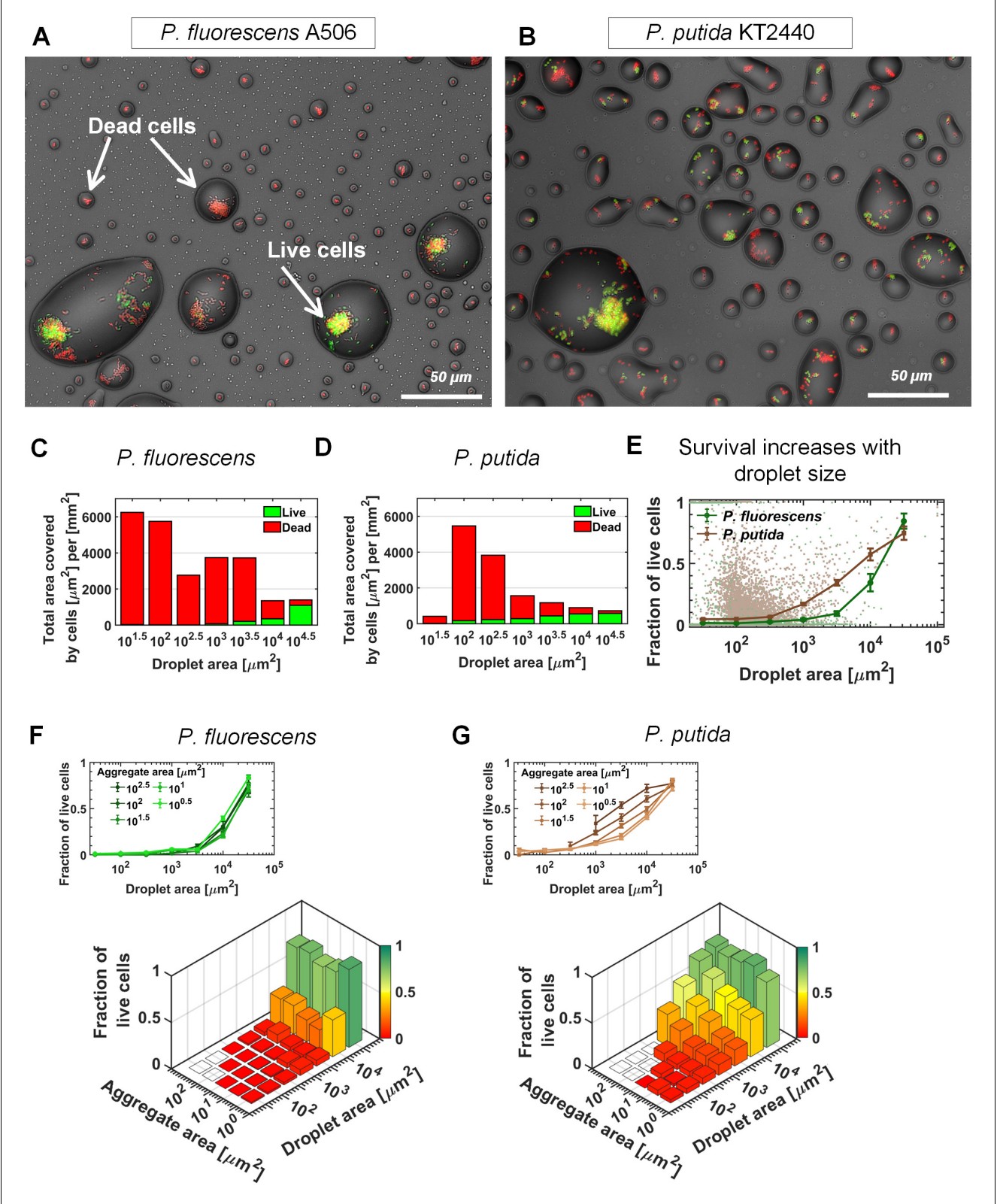

**Figure 3.** Bacterial survival increases with droplet size. (A–B) A section of the surface covered with droplets (experiment with *P. fluorescens* (A) and *P. putida* (B), 24 hr after macroscopic drying): Live cells are green, and dead cells (cells with damaged membrane) are red. Live cells were mostly observed in large droplets. (C–D) *P. fluorescens* (C) and *P. putida* (D) cell distributions, binned by droplet size: The green and red colored bars indicate the fraction of live and dead cells respectively. (E) Fraction of live cells as a function of droplet size. Survival rate increases with droplet size in both studied

*Figure 3 continued on next page*

*Figure 3 continued*

strains. Error bars represent standard errors. Each dot in the background represents a single droplet. (F) *P. fluorescens* survival rates as a function of aggregate size and droplet size: The height of the bars indicates all cellular-object (aggregates or solitary cells) mean survival rates within a given bin of aggregate and droplet size. The inset above shows the same data, but presented differently, with each line representing an aggregate-size bin. Note that there is no pronounced difference between lines, indicating that aggregate size has only minor effect on *P. fluorescens* survival. (G) Same as (F) but for *P. putida*: Note the pronounced difference between lines, indicating that aggregate size contributes to *P. putida*'s survival (larger aggregates have higher survival); yet droplet size contributes to survival more profoundly than does aggregation (see also *Figure 3—figure supplement 3*).

DOI: https://doi.org/10.7554/eLife.48508.015

The following source data and figure supplements are available for figure 3:

**Source data 1.** Survival rates and their relation to droplet and aggregate size.
DOI: https://doi.org/10.7554/eLife.48508.024
**Figure supplement 1.** Viability of cells within microdroplets.
DOI: https://doi.org/10.7554/eLife.48508.016
**Figure supplement 2.** Survival as a function of aggregate size.
DOI: https://doi.org/10.7554/eLife.48508.017
**Figure supplement 2—source data 1.** Mean survival and standard error of survival rates within aggregates, binned by aggregate area.
DOI: https://doi.org/10.7554/eLife.48508.018
**Figure supplement 3.** Multinomial logistic regression model.
DOI: https://doi.org/10.7554/eLife.48508.019
**Figure supplement 4.** Survival rate of solitary cells increases with droplet size.
DOI: https://doi.org/10.7554/eLife.48508.020
**Figure supplement 4—source data 1.** Mean survival and standard error of survival rates within droplets, binned by host droplet area.
DOI: https://doi.org/10.7554/eLife.48508.021
**Figure supplement 5.** Growth curves of *P. fluorescens* A506 and *P. putida* KT2440 under various M9 dilutions/concentrations and under various NaCl concentrations.
DOI: https://doi.org/10.7554/eLife.48508.022
**Figure supplement 5—source data 1.** Growth curves for *P. fluorescens and P. putida* strains under various M9/NaCl concentrations.
DOI: https://doi.org/10.7554/eLife.48508.023

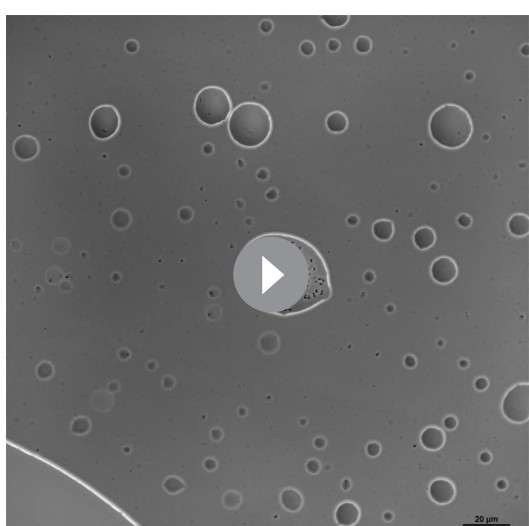

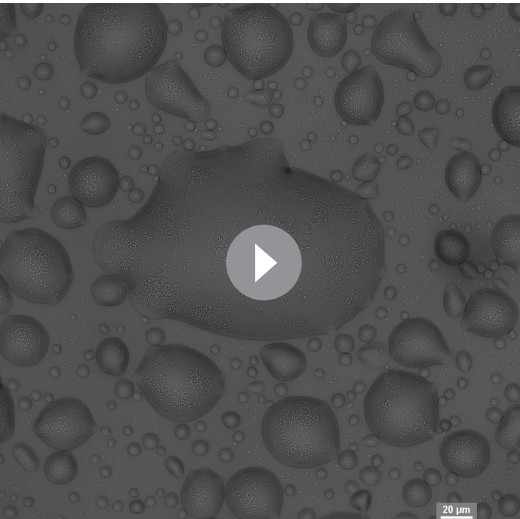

**Video 4.** Viability of cells within a microdroplet. Some *P. fluorescens* cells can be seen swimming, confined within the droplet. Cells were recovered for 84 hr at 95% RH after our standard surface drying experiment (85% RH). The video plays at real-time speed.
DOI: https://doi.org/10.7554/eLife.48508.025

**Video 5.** Viability of cells within a microdroplet. Some *P. fluorescens* cells can be seen swimming, confined within the droplet. Cells were recovered for 48 hr at 95% RH after our standard surface drying experiment (85% RH). The video plays at real-time speed.
DOI: https://doi.org/10.7554/eLife.48508.026

– so that the cells did not have time to grow and form aggregates, and were thus mostly solitary. Notably, live cells were observed nearly exclusively in large droplets (>10$^3$ μm$^2$ area, cf. *Figure 3—figure supplement 4*), and survival increased with droplet size. These results indicate that large droplets promote cell survival even when aggregates are absent.

Experiments with 16 additional strains, including Gram-negative and Gram-positive bacteria from a variety of microbial habitats, yielded qualitatively similar results to those described in the preceding paragraphs (*Table 1*). Although not all the strains formed aggregates under our experimental conditions, the general picture was same for all strains: Larger droplets were observed around aggregates or surface areas more densely populated by cells (for strains that did not form aggregates), and higher survival was observed in larger droplets.

## Formation of droplets using dissolved solutes and microbiota from natural leaves

Lastly, we repeated our experiments using solutes and microbiota extracted from the surface of a natural leaf. We found that stable microdroplets also formed around natural microbiota cells, in some cases only at higher RH (>85%) or at lower temperatures, suggesting that condensation is involved in microdroplet formation. Furthermore, the microscopic wetness from natural leaf wash was visibly similar to those in our experiments with inoculated bacteria and a synthetic medium (*Figure 4*, *Figure 4—figure supplement 1*).

## Discussion

Our study demonstrates that stable microdroplets of concentrated liquid solutions form around cells and aggregates on bacterial-colonized surfaces that are drying under moderate to high RH. We show that bacterial cell organization on a surface strongly affects the microscopic hydration conditions around cells, and that droplet size strongly affects cell survival. We reveal an additional function

**Table 1.** Strains used in this study.

'Aggregation' was determined as 'yes' if the majority of cells (>~50%) were observed in clusters of more than five individual cells, and 'no' otherwise. Survival level was estimated as follows: 'low': almost no survival (<3% of cells); 'medium': survival of 3% to 50% of the cells; 'high':>50% of all cells survived.

| Genus | Species | Strain | Gram +/- | Major habitat | Aggregation | Survival at 24 hr | Gifted from |
|---|---|---|---|---|---|---|---|
| *Pseudomonas* | *syringae* | B728a | - | phyllosphere | No | low | S. Lindow |
| *Pseudomonas* | *syringae* | DC3000 | - | phyllosphere | No | low | O. Bahar |
| *Pseudomonas* | *fluorescens* | A506 | - | phyllosphere | Yes | medium | S. Lindow |
| *Pseudomonas* | *fluorescens* | NT133 | - | rhizosphere | Yes | low | D. Minz |
| *Pseudomonas* | *putida* | KT2440 | - | soil | Yes | medium | Purchased from ATCC |
| *Pseudomonas* | *putida* | KT2442 | - | soil | Yes | medium | Y. Friedman |
| *Pseudomonas* | *putida* | IsoF101 | - | soil | Yes | medium | L. Eberl |
| *Pseudomonas* | *citronellolis* | 13674 (ATCC) | - | soil | Yes | low | Y. Friedman |
| *Pseudomonas* | *aurantiaca* | 33663 (ATCC) | - | soil | Yes | medium | Y. Friedman |
| *Pseudomonas* | *veronii* | 700474 (ATCC) | - | water | Yes | high | Y. Friedman |
| *Pantoea* | *agglomerans* | 299 r | - | phyllosphere | No | low | S. Lindow |
| *Pantoea* | *agglomerans* | BRT98 | - | soil | Yes | high | Z. Cardon |
| *Escherichia* | *coli* | K-12 MG1655 | - | human gut | No | medium | Y. Helman |
| *Xanthomoas* | *campestris* | 85–10 | - | phyllosphere | No | low | G. Sessa |
| *Burkholderia* | *cenocepacia* | H111 | - | human | Yes | low | Y. Helman |
| *Acidovorax* | *citrulli* | M6 | - | phyllosphere | No | low | S. Burdman |
| *Bacillus* | *subtilis* | 3610 | + | soil | No | low | Y. Helman |
| *Clavibacter* | *michiganensis* | | + | soil | Yes | low | S. Burdman |

DOI: https://doi.org/10.7554/eLife.48508.027

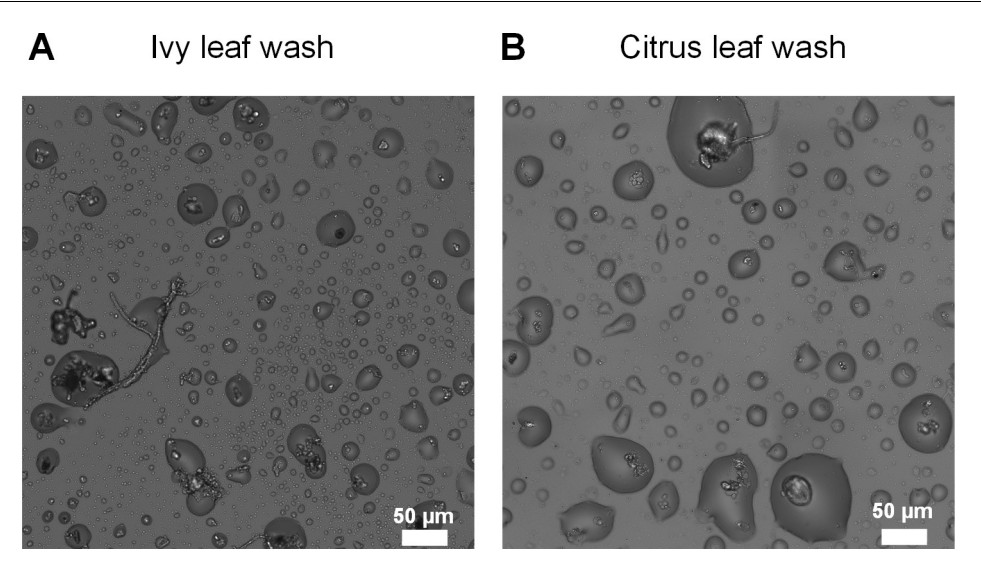

**Figure 4.** Microscopic wetness forming with natural leaf washes. (**A**) ivy leaf wash. (**B**) orange leaf wash. In both leaf washes, droplet formation around microbiota cells including fungi, yeast, and bacteria can be observed. Leaf wash protocols and drying conditions are described in Materials and methods.

DOI: https://doi.org/10.7554/eLife.48508.028

The following figure supplement is available for figure 4:

**Figure supplement 1.** Microscopic surface wetness forming with natural leaf washes.

DOI: https://doi.org/10.7554/eLife.48508.029

of bacterial aggregation: improving hydration by retaining large stable droplets (>tens of μm in diameter) around aggregates. Why survival is enhanced in larger droplets remains an open question. We hypothesize that larger droplets provide favorable conditions due to higher water potential; further research is required to test this hypothesis.

We note that the evaporation dynamics of a drop of a liquid solution – even without bacteria – is a surprisingly rich and complex physical process and a subject of intensive research (*de Gennes et al., 2013*; *Bonn et al., 2009*). Our results point to two central mechanisms promoting the formation and stability of microdroplets around bacterial aggregates: The first is pinning of the liquid-air interface due to the large interfacial tension force associated with the rough surfaces of particulate aggregates (*Herminghaus et al., 2008*; *Bonn et al., 2009*), as observed in *Videos 1–3*. The second is the deliquescent property of solutes that prohibits complete evaporation of the pinned droplets at RH that is higher than the point of deliquescence of the solutes, such that the droplets are in equilibrium with the surrounding humid air.

We suggest that bacterial self-organization on surfaces can improve survival in environments with recurrent drying that lead to microscopic wetness. A simple conceptual model that captures the system's main components and their interactions is depicted in *Figure 5A*. Aggregation is an important feature that can affect self-organization, and in turn, the resulting waterscape, by increasing the fraction of the population that ends up in large droplets. Preliminary evidence for this is provided by the comparison of the fraction of the population residing in droplets above a given size, using beads, 'solitary' and 'aggregated' cells as particles (*Figure 5B*, *Figure 5—figure supplement 1*). The interplay between self-organization, waterscape, and survival is an intriguing open question that merits further research.

Interestingly, the ecological origin of the strains (*Table 1*) did not always predict their survival rates. Some phyllospheric bacteria (mostly plant pathogens) exhibited low survival, soil bacteria exhibited variable survival rates, *E. coli* exhibited a surprising medium survival, and the aquatic strain *P. veronii* exhibited high survival. Survival in microscopic surface wetness is likely a complex trait that

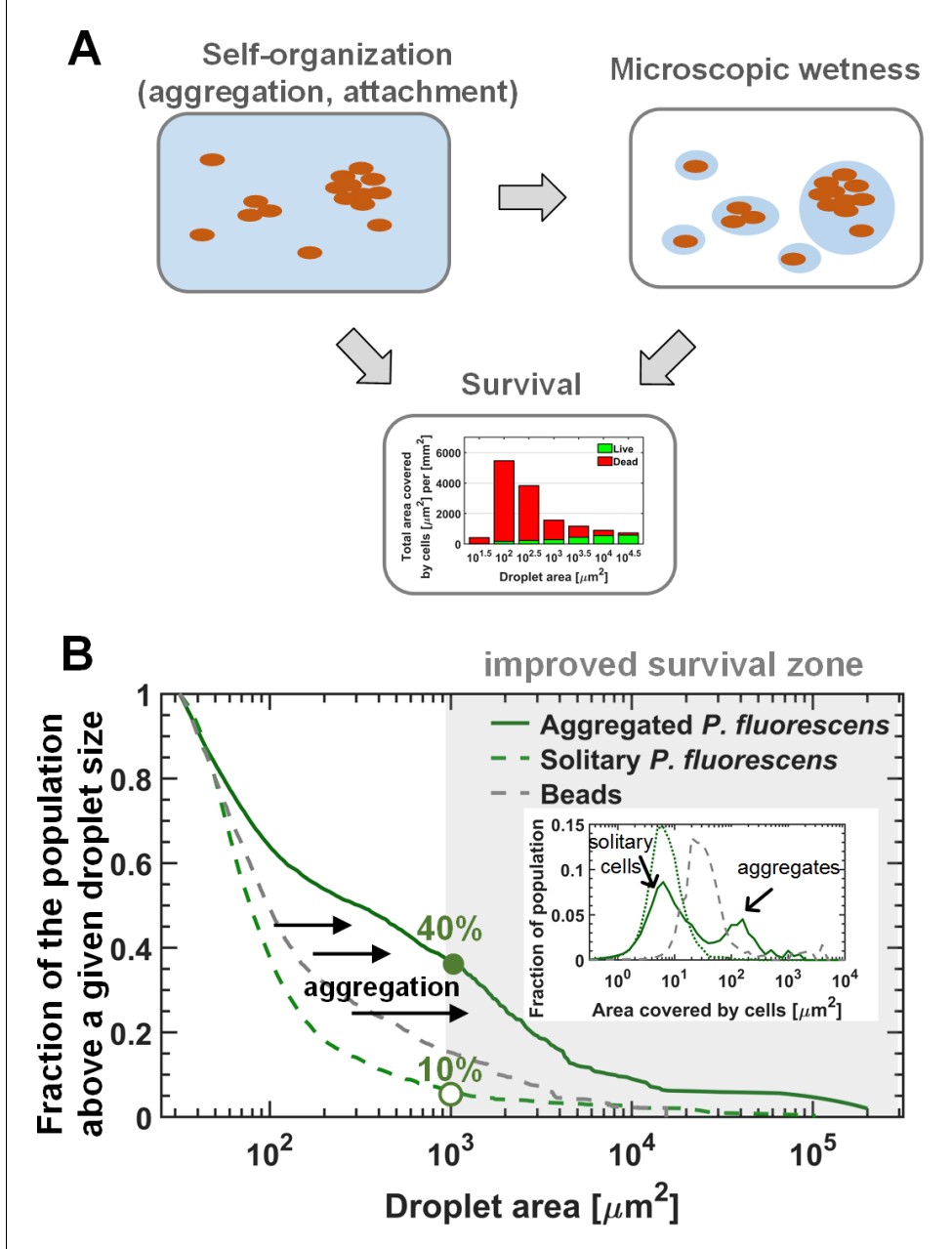

**Figure 5.** The interplay between self-organization, waterscape, and survival. (**A**) Suggested conceptual model: The self-organization of cells on the surface affects the microscopic waterscape and the microscopic hydration conditions around cells, which in turn, together with cellular organization (i.e. aggregation) affects survival. (**B**) The three lines represent the fraction of the population residing above a given droplet size (i.e. the ratio between the area covered by cells residing in droplets larger than a given size, to the total area covered by cells) of the solitary (late-inoculation) experiment, the bead experiment, and the standard "aggregated "experiment on *P. fluorescens*. Inset: Aggregate-size distributions of these three experiments. Aggregation results in a larger fraction of the population ending up in large droplets with increased survival rates for cells therein.

DOI: https://doi.org/10.7554/eLife.48508.030

The following source data and figure supplements are available for figure 5:

**Source data 1.** *Figure 5B*: Fraction of cells or beads(estimated by area) residing above a given droplet size.
DOI: https://doi.org/10.7554/eLife.48508.033
**Figure supplement 1.** Aggregation and self-organization affect survival.
DOI: https://doi.org/10.7554/eLife.48508.031

*Figure 5 continued on next page*

*Figure 5 continued*

**Figure supplement 1—source data 1.** Fraction of the population residing above a given droplet size.
DOI: https://doi.org/10.7554/eLife.48508.032

combines physiological adaptation of the individual cell and collective protection that results from self-organization and cooperation (i.e. aggregation). In nature, bacteria live in complex communities comprised of many bacterial species and are exposed to various chemical and physical environmental cues. Thus, our single-strain experiments, with M9 medium on glass-bottom wells, may not capture survival strategies that might be triggered by environmental cues and that rely on other members of the community. For example, joining existing aggregates of other species can be a beneficial strategy in environments with recurrent drying events (*Grinberg et al., 2019*; *Steinberg et al., 2019*).

Obviously, there are more differences between natural leaf surfaces and our simplified experimental system. Firstly, leaf surfaces have heterogeneous 3D topography due to leaf microscale anatomy such as the cavities between epidermal cells, stomata openings, and trichomes (*Koch et al., 2008*). This microscale topography affects drying and wetting of the leaf surface, and hence can impact droplet formation both by its effect on interfacial forces and pinning as well as imposing stronger flow upon topological sinks. Secondly, leaf surfaces tend to be hydrophobic to a degree that varies among plant species (*Koch et al., 2008*). The impact of both microscale topography and surface hydrophobicity on drying and droplet formation can be studied using artificial leaves (*Doan and Leveau, 2015*; *Soffe et al., 2019*) or leaf cuticle peels (*Schönherr and Riederer, 1986*; *Remus-Emsermann et al., 2011*). Finally, the chemical composition of leaf surface wetness varies considerably with multiple factors including plant species, soil characteristics (e.g. salinity), geography, and environmental variables that affect atmospheric aerosol composition, deposition, and retention, such as wind and rain (*Pöschl, 2005*; *Tang and Munkelwitz, 1993*; *Tang, 1979*). All of these factors are likely to affect the formation and retention of microscopic leaf wetness.

Our results suggest that microscopic surface wetness, predicted to occur globally on plant leaves (*Burkhardt and Hunsche, 2013*), can explain how microorganisms survive on leaf surfaces during daytime by avoiding complete desiccation. Yet, they also imply that phyllospheric bacteria have evolved mechanisms to cope with the highly concentrated solutions associated with deliquescent wetness. The ability to tolerate periods of such high salinities could thus be a ubiquitous and necessary trait for phyllospheric bacteria. Better understanding of bacterial survival in microscopic deliquescent surface wetness, and how it is affected by agricultural practices and anthropogenic aerosol emissions, is thus of great importance to microbial ecology of the phyllosphere and to plant pathology.

Finally, as deliquescent substances are prevalent in many other microbial habitats, it is safe to assume that deliquescent microscopic wetness occurs in many microbial habitats, including soil and rock surfaces (*Davila et al., 2008*; *Davila et al., 2013*), the built environment, human and animal skin, and even extraterrestrial systems (e.g. Mars; *Nuding et al., 2017*; *Stevens et al., 2019*). Moreover, microscopic surface wetness is likely to have a significant impact not only on survival, but also on additional key aspects of bacterial life, including motility, communication, competition, interactions, and exchange of genetic material, as demonstrated for soil and other porous media (*Tecon et al., 2018*; *Or et al., 2007*). Microbial life in deliquescent microscopic surface wetness remains to be further explored.

# Materials and methods

**Key resources table**

| Reagent type (species) or resource | Designation | Source or reference | Identifiers | Additional information |
|---|---|---|---|---|
| Strain, strain background (*Pseudomonas fluorescens*) | A506 | DOI: 10.17660/ActaHortic.1993.338.51 | | Gifted from S. Lindow lab |
| Strain, strain background (*Pseudomonas putida*) | KT2440 | ATCC | 47054 | |
| Recombinant DNA reagent | pUC18T-mini-Tn7T-Gm-eyfp (plasmid) | Addgene | 65031 | |
| Recombinant DNA reagent | pTNS1 (plasmid) | Addgene | 64967 | |
| Software, algorithm | NIS Elements 5.02 | Nikon Instruments | RRID:SCR_014329 | |
| Software, algorithm | MATLAB | MathWorks | RRID:SCR_001622 | |
| Other | propidium iodide stain | Invitrogen | L-7012 | component B, LIVE/DEAD Bac-Light Bacterial Viability Kit |
| Other | fluorescent beads | FLUKA | 94009 | rhodamine-tagged micro particles (2 μm) |
| Other | Alexa Fluor 647 Carboxylic Acid, tris (triethylammonium) salt | Invitrogen | A33084 | |
| Other | Sticker; SecureSeal Imaging spacers | Grace Bio-Labs | SS1 × 20 | |

## Experimental design

A simple experimental system, accessible to microscopy, that enables studying the interplay between bacterial surface colonization, cell survival, and microscopic wetness on artificial surfaces was built (see *Figure 1B* and section Drying surface experiments). Fluorescently tagged bacterial cells are inoculated in liquid media onto hollowed stickers adhered to the glass substrate of multi-well plates and placed inside an environmental chamber under constant temperature and RH (*Figure 1B*, and Drying surface experiments and Strains and culture condition). After macroscopic drying is achieved, plates are examined under the microscope (see Microscopy) and microscopic wetness, bacterial surface colonization, and cell survival are analyzed (see Image analysis, Statistical analysis, and Estimation of solution concentrations within droplets). Similar experiments with natural leaf washes are described in the section Natural leaf washes.

## Drying surface experiments

Imaging spacers (20 mm SecureSeal SS1 × 20, Grace Bio-Labs) were used to confine the inoculum on the surface of six-well glass bottom plates (CellVis) (*Figure 1B*). The spacer was used to reduce flow dynamics effect that result in transfer of biomass to the edge of an evaporating body of liquid drops on flat surfaces (e.g. coffee ring effect; *Deegan et al., 1997*; *Larson, 2017*). Reduction of flow was achieved through a more spatially uniform evaporation rate. The corners of the spacer were cut to fit the well, adhesive liner was removed from one side of the spacer, and the exposed adhesive was applied to the center of the well by applying gentle pressure against the glass using a sterile disposable cell spreader. The upper liner was removed and the hollow of the spacer was loaded with 340 μl of diluted suspended cells (~$2\times10^3$ cell/ml) at half-strength M9 medium (with 2 mM glucose conc.). For survival assay, propidium iodide (component B, LIVE/DEAD Bac-Light Bacterial Viability Kit, L-7012, Molecular Probes) was added to the starting inoculum to obtain a final concentration of 20 nM. The typical 'live' SYTO dye was not used; instead, we used the constitutive YFP expression of

live cells (see below) as indication of living cells. In the experiments with fluorescent beads, rhodamine-tagged micro particles (2 μm) based on melamine resin were used (melamine-formaldehyde resin, FLUKA). The plates were placed, with the plastic lid open, on the uppermost shelf of a temperature- and humidity-controlled growth chamber (FitoClima 600 PLH, Aralab). Temperature was set to 28°C, RH to 70% or 85%, and fan speed to 100%. Prior to the microscopy imaging acquisition, diH$_2$O was added to the empty spaces between the wells of the plate, plates were covered with the plastic lid, and the plate's perimeter was sealed with a stretchable sealing tape to maintain a humid environment (>95% RH).

## Bacterial strains and culture conditions

*Pseudomonas fluorescens* A506 (*Wilson and Lindow, 1993*; *Hagen et al., 2009*) and *Pseudomonas putida* KT2440 (*Nelson et al., 2002*) (ATCC 47054) were chromosomally tagged with YFP using the mini-Tn7 system (*Choi and Schweizer, 2006*) (Plasmid pUC18T-mini-Tn7T-Gm-eyfp and pTNS1, Addgene plasmid # 65031, and # 64967 respectively (*Choi et al., 2005*). Prior to the gradual drying experiments, strains were grown in LB Lennox broth (Conda) supplemented with gentamicin 30 μg/ml for 12 hr (agitation set at 220 rpm; at 28°C). 50 μl of the 12 hr batch culture was transferred into 3 ml of fresh LB medium, and incubated for an additional 3–6 hr (until OD reached a value of ~0.5–0.7). Suspended cells were transferred to a half-strength M9 medium supplemented with glucose by a two-step washing protocol (centrifuge at 6000 rcf for 2 min., and resuspension of the pellet in 500 μls medium). The half-strength M9 medium consisted of 5.64 g M9 Minimal Salts Base 5x (Formedium), 60 mgs of MgSO4, and 5.5 mgs of CaCl2 per liter of de-ionized water supplemented with 360 mgs glucose as a carbon source (final glucose concentration of 2 mM). The full list of strains used in this study is given in *Table 1*.

## Microscopy

Microscopic inspection and image acquisition were performed using an Eclipse Ti-E inverted microscope (Nikon) equipped with 40x/(0.95 N.A.) air objective. A LED light source (SOLA SE II, Lumencor) was used for fluorescence excitation. YFP fluorescence was excited with a 470/40 filter, and emission was collected with a T495lpxr dichroic mirror and a 525/50 filter. Propidium iodide fluorescence was excited with a 560/40 filter, and emission was collected with a T585lpxr dichroic mirror and a 630/75 filter (filters and dichroic mirror from Chroma). A motorized encoded scanning stage (Märzhäuser Wetzlar GmbH) was used to collect multiple stage positions. In each well, five xy positions were randomly chosen, and 5 × 5 adjacent fields of view (with a 5% overlap) were scanned. Images were acquired with an SCMOS camera (ZYLA 4.2PLUS, Andor). NIS Elements 5.02 software was used for acquisition and basic image processing.

## Image analysis

The images were exported from NIS Elements as four separate 16-bit grayscale images per image: bright field (BF), YFP fluorescence (green), propidium fluorescence (red), and a shorter wavelength fluorescence that highlights the droplets (blue). Image analysis was performed in MATLAB. The droplets were segmented by processing the blue fluorescence channel. Droplets were segmented by setting thresholds on the image intensity and gradient following Gaussian filtering (the centers of the droplets are brighter than their periphery and background, and the gradient is more pronounced at the periphery). The two resulting masks were combined, and holes in the connected components were removed. Live and dead cells within each droplet were segmented by the histogram-based threshold of the green and red fluorescent channels respective intensities, producing binary segmentation and live/dead classification of the cells. The segmented droplet image was then used to assign cells and aggregates to their 'host' droplet, and to quantify the live/dead surface coverage within each droplet and aggregate.

Our analysis relies on the projected 2D features of 3D objects: droplets and bacterial cells and aggregates. Although some information is lost in the projection, it was deemed a necessary tradeoff for the analysis of the large scanned area and the quantity of data involved. We assume that the relationship between droplet area and volume is monotonous, and that the great majority of cellular aggregates are single layered. To affirm these assumptions, we performed 3D analysis using z-stacking and 3D deconvolution on a small surface area. This analysis verified that our droplet

identification and segmentation does not capture flat discolorations as droplets, and that indeed the cells within the droplets are generally arranged in a single layer on the surface, or suspended in the liquid at densities low enough to maintain the validity of 2D projections.

## Statistical analysis

Data analyses and statistics for experiments with bacterial cells were based on microscopy images of five different surface sections (each of an area of 2.5 mm$^2$) per well. Data analyses and statistics for experiments with beads were based on microscopy images of surface sections of areas of 10 mm$^2$. For statistical analysis of mean values and standard errors, droplets and aggregates were binned by their size on a logarithmic scale. In *Figure 2C*, standard errors are based on the five surface sections (of 2.5 mm$^2$) per strain (n = 5) and nine different surface sections (of 1.1 mm$^2$) for the beads experiment (n = 9). In *Figure 2D* and *Figure 3E*, standard errors are calculated for all droplets within each bin (size range of droplets) of the combined data of the five surface sections for experiments with bacteria. In *Figure 3B,C* and *Figure 4B*, data is combined for all five surface sections. In *Figure 3F, G* standard errors are calculated for all aggregates within each bin of the combined data of the five surface sections.

## Estimation of solution concentrations within droplets

### Calibration curves

Concentrated stock solutions of: (1) M9 salts 100x, Alexa Fluor 647 dissolved in diH$_2$0 were prepared (M9 minimal salts base, 5x, ForMedium; sodium chloride, J.T.Baker; Alexa Fluor 647 carboxylic acid, tris (triethylammonium) salt, Invitrogen), and (2) NaCl 4M, Alexa Fluor 647 100 µM in diH$_2$0. In order to build concentration calibration curves (i.e., a graph that describes the fluorescence intensity versus known concentration of M9 or NaCl solutions), the stock solutions were diluted (in diH$_2$O) by the following factors: 1.11, 1.25, 1.43, 1.6, 2, 2.5, 3.33, 5, 10 and 20. A 5 µl drop was pipetted out of each diluted sample and placed on the glass surface (thickness 0.15 mm) of a 24-well plate that was pre-equilibrated with a reservoir of tap water to maintain humid conditions. Drops were imaged by confocal microscopy.

### Droplets concentration

Droplets from M9 (0.5x and 0.05x) and NaCl (16 mM, 40 mM) solutions were formed through our standard drying surface experiments with 2 µm beads, as described previously. Droplets were imaged by confocal microscopy 18 hr after the plates were placed in the growth chamber (28˚C, 85% RH).

### Confocal microscopy

Confocal imaging was acquired using a LEICA SP8 (CTR6000) microscope with a Leica HC PL APO CS2 40x (1.10 N.A.) water objective. Lasers line 638 nm was used for excitation of Alexa 647, and emission was collected between 656–684 nm. All images were collected with a PMT detector (laser intensity 0.01 or 0.05 and Gain 680 or 640 for M9 or NaCl calibration curves, respectively).

### Image processing and data analysis

The calibration curves were obtained by measuring the mean intensity of the 5 µl drops of known concentrations, for M9 and NaCl separately. For each solute fixed resolution, laser intensity and photomultiplier gain values were used. The intensity of the drops was defined as the mean intensity of 1,000 × 1,000 pixels (109 µm x 109 µm area) within the drop, without overlapping with the drop edges. The series of intensity-concentration pair data points were used to build the calibration curves, by piecewise-linear interpolation. The concentrations of microdroplets from the drying surface experiments were estimated by measuring their intensity and converting this value to concentrations using the calibration curves. Microdroplet intensity was measured by imaging a 520 µm x 520 µm square with the same resolution, laser intensity, and photomultiplier gain values as for the 5 µl drops of the respective solute (M9 or NaCl). The microdroplets were segmented by setting a threshold value higher than that of the background intensity. The intensity value for each microdroplet was calculated by averaging the intensity of the entire microdroplet, excluding a ~ 1 µm-wide boundary, and the area occupied by the beads (determined by BF intensity threshold).

### Natural leaf washes

We employed two different methods for the extraction and drying of leaf washes.

## Method I

A single ivy and orange leaf were submerged in separate sterile petri dishes filled with 10–15 mL autoclaved diH$_2$O. The leaves were gently rubbed while submerged to remove particles on the leaf surface. 340 µl of the leaf wash solution was loaded into a sticker in a six-well plate and dried overnight in a growth chamber at 28°C and 85% RH. After macroscopic drying, water was added to the spacers in between the wells, the plate was sealed with tape and returned to incubation at 28°C and ~95% RH. Images in *Figure 4* are after 48 hr of incubation. This method was used to obtain images in *Figure 4*.

## Method II

Several 200 µl diH$_2$0 drops were loaded onto the abaxial surface of an ivy leaf, and left for 2 hr (room temperature). A volume of 340 µl sampled from several drops was aspired with a pipette, transferred to our standard surface drying platform, and left to dry using our described protocol (28°C, 85% RH). This method was used to obtain images in *Figure 4—figure supplement 1*.

## Acknowledgements

We thank Y Helman, Y Friedman, Y Hadar, E Jurkevitch, O Yarden, R Holtzman, O Bäumchen, D Sher, A Bren, and S Itzkovitz for valuable comments and discussions. We thank S Lindow, L Eberl, Z Cardon, D Minz, O Bahar, G Sessa, S Burdman, Y Helman, and Y Friedman, for kindly providing bacterial strains. This work was supported by research grants to NK from the James S McDonnell Foundation (Studying Complex Systems Scholar Award, Grant #220020475) and from the Israel Science Foundation (ISF #1396/19).

## Additional information

### Funding

| Funder | Grant reference number | Author |
|---|---|---|
| James S. McDonnell Foundation | #220020475 | Nadav Kashtan |
| Israel Science Foundation | 1396/19 | Nadav Kashtan |

The funders had no role in study design, data collection and interpretation, or the decision to submit the work for publication.

### Author contributions

Maor Grinberg, Conceptualization, Data curation, Software, Formal analysis, Investigation, Visualization, Methodology, Writing—original draft, Writing—review and editing; Tomer Orevi, Conceptualization, Resources, Formal analysis, Investigation, Visualization, Methodology, Writing—original draft, Writing—review and editing, Performing experiments; Shifra Steinberg, Investigation, Writing—review and editing, Performing experiments; Nadav Kashtan, Conceptualization, Supervision, Funding acquisition, Investigation, Visualization, Writing—original draft, Writing—review and editing

### Author ORCIDs

Nadav Kashtan (iD) https://orcid.org/0000-0002-7475-1363

### Decision letter and Author response

Decision letter https://doi.org/10.7554/eLife.48508.038
Author response https://doi.org/10.7554/eLife.48508.039

# Additional files

## Supplementary files

• Supplementary file 1. Molar concentration and osmolarity of M9 salts. Calculated molarity and osmolarity of salt components in M9 medium at the standard concentration (M9 1x) and in estimated concentrations within microdroplets (M9 23.3x).

DOI: https://doi.org/10.7554/eLife.48508.034

• Supplementary file 2. Growth curve analysis of *P. fluorescens* A506 and *P. putida* KT2440 at various M9 concentrations and NaCl concentrations. Plate Reader (Synergy H1, BioTek) screen results were analyzed using GrowthRate and GRplot programs (Mira, P., M. Barlow, and B. G. Hall. Statistical Package for Growth Rates Made Easy. Mol. Biol. Evol. 34:3303–3309, 2017). Results of zero growth were omitted from this table. In both strains, the general picture was that higher salt concentrations led to a decrease in growth rate, a decrease in final OD, and an increase in lag time. '*': R is lower than 0.99.

DOI: https://doi.org/10.7554/eLife.48508.035

• Transparent reporting form

DOI: https://doi.org/10.7554/eLife.48508.036

## Data availability

All data generated or analysed during this study are included in the manuscript and supporting files. Source data files have been provided for Figures 2, 3 and 5.

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
