## [Decision Letter]

Thank you for submitting your article "Bacterial survival in microscopic droplets" for consideration by *eLife*. Your article has been reviewed by four peer reviewers, including Wenying Shou as the Reviewing Editor and Reviewer #1, and the evaluation has been overseen by Gisela Storz as the Senior Editor. The following individual involved in review of your submission has agreed to reveal their identity: Robin Tecon (Reviewer #3).

The reviewers have discussed the reviews with one another and the Reviewing Editor has drafted this decision to help you prepare a revised submission.

Summary:

In this study, the authors investigate how physical, microhydrological conditions affect bacterial populations on a drying surface, and attempt to mechanistically link bacterial density, droplet size, and bacterial survival. Using artificial surfaces and systematic control of evaporation conditions, they demonstrate that bacterial cells and cell aggregates can retain water in the form of microdroplets invisible to the naked eye. The size of droplets positively correlates with the number of trapped cells as well as the probability of cell survival. Results were relatively consistent across a dozen bacterial species and with natural leaf communities, as well as on two types of surfaces and two values of relative humidity.

Essential revisions:

1) A more explicit comparison between their system and the surface of a natural leaf.

Reviewer 3 suggested: "leaf surface mimicking comes from creating a transition from saturated (wet, dew at night) to unsaturated (microscopic droplets persisting on dry surface but invisible to the naked eye). This is possible thanks to controlled evaporation at high RH. The authors could discuss more how this relates to the leaf surface 'microclimate'. Moreover, they tested natural leaf washes (using bidistilled water), which match the chemical and biological composition of leaf surface."

"They could perform more experiments with more complex surfaces, e.g. using artificial leaf surfaces, or leaf cuticle peeled from leaf (see Bisha and Brehm-Stecher, AEM 2009). However, I think that such experiments with leaf surfaces are (i) not essential to the general conclusions drawn by the authors, and (ii) would very likely require more than 2 months’ work." Perhaps, add a section on future directions in the Discussion.

2) Better characterize salt concentrations in droplets of different sizes and their effects on bacterial growth and survival (including maybe in natural leaf washes).

3) Explain better the physical (hydrological) processes at play in the Discussion.

4) It would be helpful to more thoroughly report/explain the results with other species reported in Table 1.

For the sake of completeness, the reviewing editor attaches abbreviated individual reviews to help make your paper stronger.

Reviewer #2:

The manuscript examines the ability of bacteria to survive in microscopic droplets that naturally form on surfaces due to a naturally occurring process called deliquescence. As a result of deliquescence, which is likely relevant to many real-world environments including leaf surfaces, tiny droplets form around bacterial cells on surfaces. The study finds that droplet size increases for bacterial aggregates, and that bacteria thrive within these larger droplets. Overall the quality of the work is very high, and the findings are both intriguing and of general interest to *eLife* readers. The Introduction does a great job justifying the approach and relevance of the study. For the most part results were clearly presented and carefully interpreted. This was one of the best papers I have read in recent years, I strongly recommend it for publication in *eLife*. I do have a few suggestions that will help improve the manuscript prior to publication.

1) The authors state that "Last, we repeated the beads experiment with pure water instead of M9 medium. This time we did not observe any droplets, indicating that the solutes control droplet formation and retention through deliquescence." I don't believe a picture of this result was included in the supplementary information. A minor point, but it would be helpful to include a picture, perhaps paired with Figure 2—figure supplement 1.

2) It would be of interest to many readers to know the approximate doubling time of cells within the droplets. Granted the growth rate should depend on the droplet size, so there will likely be multiple growth rates, but an estimate would be appreciated for at least some of the larger droplets with the most growth.

3) The last paragraph of the subsection “Cell survival rate increases with droplet size” is very confusing. I was unsure of the meaning "inoculated the cells at a later stage". Cells did not have time to form aggregates? Wouldn't some aggregates form by chance upon adding cells to the plate? Some clarification would be helpful.

4) A few graphs, including Figure 4B, plot "fraction of population", which seems to mean "fraction of the population residing above a given droplet size". Could you clarify the meaning or describe how it was calculated? Is the definition of population the same for Figure 4B and the inset of Figure 4B?

5) The authors state, "also imply that phyllospheric bacteria have evolved mechanisms to cope with the highly concentrated solutions associated with deliquescent wetness". This seems rather speculative and also points to one weakness of the paper. How do the concentrations of media components in the deliquescent droplets compare with the starting M9 media? Is the concentration of media components 2X, 5X? Is there a way to estimate to measure these concentrations in a droplet? At what concentration of media would cell growth be affected? It might be an helpful addition to the SI to run an additional experiment to measure growth in different concentrations of M9 (not required but would be helpful).

6) Testing many bacterial strains is an excellent addition to the paper. It would be helpful to clarify the results with other strains. How was aggregation yes or no determined? What does survival at 24h low, medium, or high indicate? These might not be very precisely defined, but some explanation should be given.

Reviewer #3:

I am quite enthusiastic about this work, and I think it is an important contribution to our understanding of the role of microhabitats in bacterial activity in natural environments. It is a field of research that is rapidly developing but that has still a lot to reveal, and which is highly complementary to research focusing on omics. Although the authors' focus is phyllosphere microbiology, as they note the research is also relevant for many other unsaturated habitats, especially soil surfaces. The authors have gathered a lot of data, which they have presented in an effective and statistically sound manner (albeit I tend to think that SD is a better descriptor than SE in such a case, but this is a detail). The fact that they tested multiple species, fluorescent beads, and different surfaces makes a good case for the generalization of the findings. Maybe for future research, the authors might consider testing artificial leaf replica (see Soffe et al., 2019; Doan and Leveau, 2015).

I have no major concerns about the study.

Reviewer #4:

The authors use model experiments to show that, when wet surfaces containing bacteria are dried, microdroplets form around bacterial aggregates, which is linked to promoting bacterial survival. The work is intriguing. However, several questions (described below) remain unanswered, and it is unclear whether this is truly an important scientific problem. I certainly would not describe this as fitting the goal of *eLife* "to publish work of the highest scientific standards and importance". The manuscript may instead be suitable after the following comments are addressed.

1) The drying experiments are performed using M9 minimal media. The authors show that the observed behavior is sensitive to the presence of solutes (by comparison to DI water drying). This casts doubt on the generalizability of the results. What is the salt content of droplets in natural settings? How do the salts in the medium (which presumably precipitate upon drying, potentially leading to e.g. droplet pinning, changes in bacterial survival, etc.) impact drying, droplet properties, and bacterial survival?

2) The authors present evidence that aggregates cause pinning and limit droplet evaporation. However, they do not provide any detailed explanation for why. What is the underlying physics? What physical forces limit the droplet evaporation? Simply showing a correlation with aggregate size does not provide any deep insight.

3) Similarly, the authors show that cell survival increases with droplet size. Why? Is this simply not due to the trivial reason that the cells have more access to more nutrients?

4) Why do aggregates form in the first place? Was this simply a consequence of incomplete mixing or improper culturing in the experiments? Can the authors control this in a way to more clearly demonstrate the influence of aggregation?

(Reviewing editor note: you can speculate in the Discussion).

---

## [Author Response]

Essential revisions:1) A more explicit comparison between their system and the surface of a natural leaf.Reviewer 3 suggested: "leaf surface mimicking comes from creating a transition from saturated (wet, dew at night) to unsaturated (microscopic droplets persisting on dry surface but invisible to the naked eye). This is possible thanks to controlled evaporation at high RH. The authors could discuss more how this relates to the leaf surface 'microclimate'. Moreover, they tested natural leaf washes (using bidistilled water), which match the chemical and biological composition of leaf surface.""They could perform more experiments with more complex surfaces, e.g. using artificial leaf surfaces, or leaf cuticle peeled from leaf (see Bisha and Brehm-Stecher, AEM 2009). However, I think that such experiments with leaf surfaces are (i) not essential to the general conclusions drawn by the authors, and (ii) would very likely require more than 2 months’ work." Perhaps, add a section on future directions in the Discussion.

We have now added a new paragraph to the Discussion that mentions the main differences between our simplified, in vitro system and natural leaves (see below). In addition, we provide new results with natural leaf washes that show microdroplet formation around indigenous leaf microbiota. We have also included a new additional figure (new Figure 4).

“Obviously, there are more differences between natural leaf surfaces and our simplified experimental system. […] All of these factors are likely to affect the formation and retention of microscopic leaf wetness.”

2) Better characterize salt concentrations in droplets of different sizes and their effects on bacterial growth and survival (including maybe in natural leaf washes).

We have invested much effort in attempts to quantify the salt concentrations in the formed microdroplets. First, we note that direct measure of salt concentrations in microdroplets of such a small volume, on a surface, is technically very challenging. To try to overcome this challenge, we tried several approaches.

Our most successful approach aimed at estimating the ‘concentration factor’, or the ratio of the concentration of solutes in microdroplets to those in the initial solution used in our drying experiments (i.e., the loaded medium with which the drying experiments commenced). To do so, we added a soluble fluorescent dye (Alexa Fluor 647) to the initial medium and compared the fluorescent intensity of the dye in the formed microdroplets to a calibration curve prepared from the fluorescence of drops with known concentration factors of the original medium. We used confocal microscopy to get good, accurate, fluorescent reads from a thin z-section of the calibration drops and microdroplets formed by our experiments.

We performed new drying experiments, with 2µm-beads as particles, and used two different mediums: M9 medium with similar composition to the one we used in our reported experiments in the paper, and a modified medium composed of just diH_2_O and NaCl. We found that the solution in droplets is 23.3 ± 3.5 (mean ± SD) times the concentration of the standard M9. That is, the final droplet solution is estimated to be ~50 times more concentrated than the original medium used in our drying experiments (an M9 χ2 dilution). We added a new paragraph summarizing these findings to the Results (“Microdroplets are highly concentrated solutions”, see below).

The details of the experiments, analyses, and results are provided in Materials and methods, and in Figure 2—figure supplements 3 and 4. The estimated concentration of the various M9 salts, as well as the estimated osmolarity (osmotic concentrations) that reflects a quantitative measurement of the overall osmolarity of the various salts, is now summarized in Supplementary file 1.

“Microdroplets are highly concentrated solutions. Direct measurements of the solute concentrations within microdroplets constitute a technical challenge. To overcome that challenge, we added a fluorescent dye (Alexa Fluor 647) to the initial M9 medium, as a reporter for the solute concentration induced by evaporation. […] This result accords with the observation that cell divisions within droplets was rarely seen in our experiments (at 85% RH).”

In addition to an M9 medium, we performed the experiment with a diH_2_O + NaCl medium and 2µm-beads. NaCl concentrations in the formed microdroplets were estimated at 650 ± 170 mM NaCl (mean ± SD), reflecting a concentrating factor of ~40 ± 10 (mean ± SD). It was interesting to find that, in the above experiments, differing initial dilutions of M9 and differing initial concentrations of NaCl reached comparable absolute concentrations in the formed microdroplets for each medium respectively. This finding accords with the predicted role of deliquescent substrates in microdroplet formation and retention. Note that (i) the point of deliquescence (POD; that is, the RH above which the salt is in liquid form) of a mixture of salts is complicated to predict and in many cases is lower than the POD of each individual salt (Tang, 1997); (ii) the difference between the surrounding RH (e.g., 85% in our experiment) and the POD of the combined salts, affects the equilibrium point and thus droplet volume and the salt concentrations therein (as can be nicely seen in our new Figure 2—figure supplement 3D-F).

Next, we tried to test whether there is a correlation between microdroplet size and the solution’s concentrations therein. Here, we could not reach a conclusive result. We saw relatively high variability in these correlations between differing fields of view with respect to micro-waterscape variations (e.g., areas that are wetter / less wet, areas dominated by films, and areas dominated by droplets). We also saw possible differences in the concentration < > area correlations between the two media we tested: the M9 and NaCl mediums. In most cases, the M9 medium showed no correlation at all, while the NaCl medium showed positive correlation. We provide these analyses and all raw data (Figure 2—figure supplements 3 and 4). However, we believe that this question requires a more thorough studythat exceeds the scope of the present paper.

Last, we performed growth curve analyses of liquid cultures of the two strains (*P. fluorescens* and *P. putida*) in two different media: (i) Various concentrations of the original M9 medium, up to the concentration range estimated for the microdroplets. (ii) A regular M9 medium with increasing concentrations of NaCl. We added a new figure to that effect: Figure 3—figure supplement 5, presenting these growth curves. We found that growth is inhibited under the estimated range of M9 concentrations within formed microdroplets (20x, 30x M9).. This accords with our observations that growth (i.e., cell division) is hardly seen in our drying surface experiments, post droplet formation. During the 24 hours since droplet formation and our survival assay, we rarely observed cell division within the droplets (at 85% RH). We did see some growth when we elevated the RH to 95%, yet still very slow growth with a division time of 23 ± 2h (mean ± SD).

An alternative approach that we have taken, to try to estimate salt concentrations in the microdroplets, was based on using bacterial whole-cell bio-reporters for osmolarity (e.g., of the proU operon). This approach has not been very successful, so far, as it appears that the various bio-reporters (including *Pantoea agglomerans, Pseudomonas putida*, and *P. fluorescens*) that we tested reached saturation at lower salt concentrations (at around ~0.5M NaCl) than did those of the solutions within microdroplets. We plan to test a bio-reporter with wider responsive range (look for one, or engineer one in our lab), for example by trying a more halophilic bio-reporter strain with reporting range that may be more suitable to our system (e.g., (Burch et al., 2013). These endeavors are still ongoing and will require further research.

3) Explain better the physical (hydrological) processes at play in the Discussion.

We expanded the relevant paragraph in the Results and added a paragraph to the Discussion that better explains the physics behind the formation and retention of microdroplets:

Results:

“To observe the surface’s final drying phase, we used time-lapse imaging, enabling us to capture the receding front of the remaining thin liquid layer and the formation of microdroplets. […] In summary, both particulates and deliquescent solutes are essential for the differential formation and retention of microscopic wetness around cells and aggregates.”

Discussion:

“We note that the evaporation dynamics of a drop of a liquid solution – even without bacteria – is a surprisingly rich and complex physical process and a subject of intensive research (De Gennes et al., 2013, Bonn et al., 2009). […] The second is the deliquescent property of solutes that prohibits complete evaporation of the pinned droplets at RH that is higher than the point of deliquescence of the solutes, such that the droplets are in equilibrium with the surrounding humid air.”

4) It would be helpful to more thoroughly report/explain the results with other species reported in Table 1.

We thank the reviewers for pointing that out. We added a more thorough summary of the results of our drying experiments with other bacterial species (see Results/Discussion changes below). In addition, we added more details to Table 1’s caption, clarifying how aggregation levels were defined and how survival levels were determined.

Results:

“Experiments with 16 additional strains, including Gram-negative and Gram-positive bacteria from a variety of microbial habitats, yielded qualitatively similar results to those described in the preceding paragraphs (Table 1). Although not all the strains formed aggregates under our experimental conditions, the general picture was same for all strains: Larger droplets were observed around aggregates or surface areas more densely populated by cells (for strains that did not form aggregates), and higher survival was observed in larger droplets.”

Discussion:

“Interestingly, the ecological origin of the strains (Table 1) did not always predict their survival rates. Some phyllospheric bacteria (mostly plant pathogens) exhibited low survival, soil bacteria exhibited variable survival rates, *E. coli* exhibited a surprising medium survival, and the aquatic strain *P. veronii* exhibited high survival. […] For example, joining existing aggregates of other species can be a beneficial strategy in environments with recurrent drying events (Grinberg et al., 2019, Steinberg et al., 2019).”

For the sake of completeness, the reviewing editor attaches abbreviated individual reviews to help make your paper stronger.Reviewer #2:[…] 1) The authors state that "Last, we repeated the beads experiment with pure water instead of M9 medium. This time we did not observe any droplets, indicating that the solutes control droplet formation and retention through deliquescence." I don't believe a picture of this result was included in the supplementary information. A minor point, but it would be helpful to include a picture, perhaps paired with Figure 2—figure supplement 1.

Good suggestion. We have added a picture of a control experiment with beads embedded in pure water to Figure 2—figure supplement 1 (side by side).

2) It would be of interest to many readers to know the approximate doubling time of cells within the droplets. Granted the growth rate should depend on the droplet size, so there will likely be multiple growth rates, but an estimate would be appreciated for at least some of the larger droplets with the most growth.

In the experimental conditions presented in our results (85% RH, 28°C) cells typically do not divide post droplet formation(at least not in droplets <10^4 µm^2^). They simply appear to survive. If RH is raised to 95%, we do then see cell recovery at least in droplets above ~ 10^3 µm^2^, including cell division, as can be seen in Figure 3—figure supplement 1. However, cell doubling time was very long. In an experiment with *P. fluorescence*, doubling time was 23 ± 2 (mean ± SD) after RH was increased to 95%.

We also added now, as mentioned in reply to Essential Revision #2, growth curve analyses at various concentrations of M9 and NaCl (see Figure 3—figure supplement 5).

3) The last paragraph of the subsection “Cell survival rate increases with droplet size” is very confusing. I was unsure of the meaning "inoculated the cells at a later stage". Cells did not have time to form aggregates? Wouldn't some aggregates form by chance upon adding cells to the plate? Some clarification would be helpful.

In our standard drying experiment, we inoculated the wells of the plate with a low density (~2x10^3^ cell/ml) of mid-log phase cells that are mostly found in planktonic state as individual cells. During the incubation period (hours), there was a gradual development of aggregates formed by clonal growth of either surface-attached cells or at the liquid-air interface (pellicles). When we inoculated the cells ~2h before the complete (macroscopic) evaporation of the medium, the cells didn’t have enough time to aggregate, at least under our experimental conditions, hence most of the cells remained solitary at the time droplets formed.

We also clarified this in the text:

“To further study droplet size’s effect on survival, we repeated the drying experiment, but inoculated the cells into the drying medium only at a later stage – closer to the macroscopic drying stage – so that the cells did not have time to grow and form aggregates, and were thus mostly solitary.”

4) A few graphs, including Figure 4B, plot "fraction of population", which seems to mean "fraction of the population residing above a given droplet size". Could you clarify the meaning or describe how it was calculated? Is the definition of population the same for Figure 4B and the inset of Figure 4B?

We now clarified this in the figure captions. The definition of population is the same for Figure 4B (new Figure 5B) and its inset.

“The three lines represent the fraction of the population residing above a given droplet size (that is, the ratio between the area covered by cells residing in droplets larger than a given size, to the total area covered by cells) of the solitary (late-inoculation) experiment, the bead experiment, and the standard “aggregated “experiment on *P. fluorescens*.”

5) The authors state, "also imply that phyllospheric bacteria have evolved mechanisms to cope with the highly concentrated solutions associated with deliquescent wetness". This seems rather speculative and also points to one weakness of the paper. How do the concentrations of media components in the deliquescent droplets compare with the starting M9 media? Is the concentration of media components 2X, 5X? Is there a way to estimate to measure these concentrations in a droplet? At what concentration of media would cell growth be affected? It might be an helpful addition to the SI to run an additional experiment to measure growth in different concentrations of M9 (not required but would be helpful).

These are excellent questions. We evaluated the salt concentrations and osmolarity in the droplets as described in detail in Essential Revision #2.

As aforementioned, we also added growth curves of the two studied stains as a function of M9 medium concentration and as a function of salt (NaCl) concentrations (Figure 3—figure supplement 5). As can be seen, growth of both strains is significantly affected at M9 medium that is 10x to 20x concentrated, which is 20x to 40x times the half strength M9 we used. Neither strain grew in 20x M9 and above. At 10x, long lag phases were observed for both strains, followed by very slow growth for *P. fluorescens* and a higher growth rate for *P. putida*. The growth curves under increasing NaCl concentrations show that the growth of *P. fluorescens* is not significantly affected up to 500mM NaCl. *P. putida* seems to be fine at high conc. of NaCl, though it can be seen that it reaches higher OD at 1,000mM NaCl only after a long lag phase. Summary of growth curve analyses are also provided in Supplementary file 2.

6) Testing many bacterial strains is an excellent addition to the paper. It would be helpful to clarify the results with other strains. How was aggregation yes or no determined? What does survival at 24h low, medium, or high indicate? These might not be very precisely defined, but some explanation should be given.

We added a clarification to Table 1 on how aggregation was determined and how survival level was determined.

Reviewer #4:The authors use model experiments to show that, when wet surfaces containing bacteria are dried, microdroplets form around bacterial aggregates, which is linked to promoting bacterial survival. The work is intriguing. However, several questions (described below) remain unanswered, and it is unclear whether this is truly an important scientific problem. I certainly would not describe this as fitting the goal of eLife "to publish work of the highest scientific standards and importance". The manuscript may instead be suitable after the following comments are addressed.1) The drying experiments are performed using M9 minimal media. The authors show that the observed behavior is sensitive to the presence of solutes (by comparison to DI water drying). This casts doubt on the generalizability of the results. What is the salt content of droplets in natural settings? How do the salts in the medium (which presumably precipitate upon drying, potentially leading to e.g. droplet pinning, changes in bacterial survival, etc.) impact drying, droplet properties, and bacterial survival?

We have conducted experiments with various mediums (M9, LB, King’s B) and with various salt compositions and concentrations (in modified M9) and at various RH levels. These results will be the basis for another study we are working on. We are quite confident regarding the generality of the results regarding the formation of microscopic surface wetness and the role of bacterial aggregates in that process (see also our reply to Essential Revision #2), especially as they are based not only on empirical observations but can also be explained by the underlying physics.

Regarding the relevance to natural systems, we believe that because deliquescent substances are ubiquitous, and moderate-to-high relative humidities – even transient ones – are very common, microscopic deliquescent-associated surface wetness likely occurs in many microbial habitats – some of them of major importance (like soil and plant leaf and root surfaces). In addition, our new results (see below) indicate that the salt concentrations in the formed microdroplets depend only weakly upon the initial salt concentrations. Thus, it is likely that microdroplet properties that matter for survival – including salt concentrations – are relatively similar across a large range of initial salt concentrations.

In nature, aerosol accumulation on leaf surface is estimated at up to 50 µg per cm^2, and higher in urban areas (Pöschl, 2005, Burkhardt and Hunsche, 2013). For example, consider a large dewdrop of 100 µLs covering a surface area of ~1 cm^2. Assuming only 1µg salt per cm^2, it reaches ~1% salts, which is approximately the percentage of salts in a standard M9 medium. As we now present in Figure 2—figure supplement 3, we found that even at lower initial concentrations of salts (e.g., diluted M9 x20 eq. to ~10mM total salts), microdroplets formed under our experimental conditions. Thus, we are quite convinced that our results are relevant to microscopic surface wetness on natural leaves.

Burkhardt and Hunsche estimated that a realistic aerosol particle load of ammonium sulfate of 5 µgs per cm^2, at 92% RH, is predicted to lead to a hypothetical ~0.5µm homogeneous water film thickness on a leaf surface (Burkhardt and Hunsche, 2013). This value is very similar to the mean thickness of the microscopic surface wetness in our experiments, considering that we typically observe that ~20% of the surface is covered by droplets of ~3µms average thickness.

To further test the relevance of our results to natural leaf surface chemistry, we conducted more experiments with natural leaf washes and natural leaf microbiota. We added to the manuscript new results with natural leaf washes (of orange and ivy plants) wherein we observed formation of stable microdroplets around natural microbiota aggregates and cells (new Figure 4).

Note that in the experiments that we conducted with M9 at 85% RH, we observed no salt precipitation. As can be clearly seen in Videos 1-3, droplet formation by pinning is tightly linked to particulates in the form of cells or beads. The resulting droplets reached an equilibrium at 85% ambient RH and did not dry out throughout the duration of the experiment. Only when exposed to a much lower RH did the droplets evaporate and the salts precipitated. To clarify this result, we added the following sentences to the relevant section of the Results:

“We note that under our experimental conditions, the droplets were not formed through the wetting ‘direction’ of a deliquescence process, by which solid salts absorb water until dissolution. […] In summary, both particulates and deliquescent solutes are essential for the differential formation and retention of microscopic wetness around cells and aggregates.”

2) The authors present evidence that aggregates cause pinning and limit droplet evaporation. However, they do not provide any detailed explanation for why. What is the underlying physics? What physical forces limit the droplet evaporation? Simply showing a correlation with aggregate size does not provide any deep insight.

Following the reviewer’s comment, we extended the relevant Results section (subsection “The underlying mechanisms of droplet formation”) and added a paragraph on the underlying physics to the Discussion (second paragraph), as described in Essential revision #3.

3) Similarly, the authors show that cell survival increases with droplet size. Why? Is this simply not due to the trivial reason that the cells have more access to more nutrients?

This is one of the key questions arising from our results. We currently do not know the answer as to why cells in larger droplets have higher survival rates. Further research is required (which we are working on). We do not believe that access to more nutrients is the factor that grants the cells higher survival. Under the experimental conditions that we tested, at 85% RH, once the droplets are formed, cells do not appear to divide at all; they appear to simply survive. We tracked cells for 40h post drying and hardly observed cell division within droplets of area < 10^4 µm^2^ for both studied strains. We therefore believe that the observed increased survival in larger droplets is related to salinity, osmotic stress, or other stresses related to water access and/or small-volume effects. pH may also vary between droplets of varying sizes, and may thus impact survival. Such significant pH differences and gradients were shown to exist even within aerosol droplets tens of microns in diameter (Wei et al., 2018).

4) Why do aggregates form in the first place? Was this simply a consequence of incomplete mixing or improper culturing in the experiments? Can the authors control this in a way to more clearly demonstrate the influence of aggregation?

Aggregation of cells was minimal at the time of adding the cells to the plate (under our experimental conditions). Flocculation was also minimal. Time-lapse imaging of the drying process, prior to macroscopic drying and droplet formation, showed that aggregates of both strains (*P. fluorescens* A506 and *P. putida* KT2440) formed mostly by clonal growth of founder surface-attached cells, and this occurred only after the addition of cells to the plate. We now clarified this in the second paragraph of the Results:

“At 85% RH, it took about 14 ± 1h for the bulk water to evaporate. During this time, for both studied strains, some of the cells attached to the surface and, over time, grew and formed aggregates. […] We then examined the surface of the wells under the microscope (see Materials and methods).”

We are also very interested in better understanding of the role of aggregation in bacterial life in microbial habitats that are characterized by microscopic surface wetness and wet-dry cycles. Indeed, we plan to study mutants that affect the cells’ tendency, or ability, to aggregate. We aim to be able to quantitatively estimate the impact of aggregation on the microscopic waterscape and in turn on fitness (e.g., survival).